# Exposure to b-LED Light While Exerting Antimicrobial Activity on Gram-Negative and -Positive Bacteria Promotes Transient EMT-like Changes and Growth Arrest in Keratinocytes

**DOI:** 10.3390/ijms23031896

**Published:** 2022-02-08

**Authors:** Michela Terri, Nicoletta Mancianti, Flavia Trionfetti, Bruno Casciaro, Valeria de Turris, Giammarco Raponi, Giulio Bontempi, Claudia Montaldo, Alessandro Domenici, Paolo Menè, Maria Luisa Mangoni, Raffaele Strippoli

**Affiliations:** 1Department of Molecular Medicine, Sapienza University of Rome, 00161 Rome, Italy; michela.terri@uniroma1.it (M.T.); flavia.trionfetti@uniroma1.it (F.T.); giulio.bontempi@uniroma1.it (G.B.); 2Gene Expression Laboratory, National Institute for Infectious Diseases, Lazzaro Spallanzani IRCCS, 00161 Rome, Italy; claudia.montaldo@inmi.it; 3Renal Unit, Department of Clinical and Molecular Medicine, Sant’Andrea University Hospital, Sapienza University of Rome, 00189 Rome, Italy; mancianti25121988@gmail.com (N.M.); alessandro.domenici@uniroma1.it (A.D.); paolo.mene@uniroma1.it (P.M.); 4Center for Life Nano- & Neuro-Science, Istituto Italiano di Tecnologia (IIT), 00161 Rome, Italy; bruno.casciaro@uniroma1.it (B.C.); valeria.deturris@iit.it (V.d.T.); 5Department of Public Health and Infectious Diseases, Sapienza University of Rome, 00185 Rome, Italy; giammarco.raponi@uniroma1.it; 6Department of Biochemical Sciences, Laboratory Affiliated to Istituto Pasteur Italia-Fondazione Cenci Bolognetti, Sapienza University of Rome, 00185 Rome, Italy; marialuisa.mangoni@uniroma1.it

**Keywords:** b-led light, skin infections, EMT, genotoxic damage

## Abstract

While blue LED (b-LED) light is increasingly being studied for its cytotoxic activity towards bacteria in therapy of skin-related infections, its effects on eukaryotic cells plasticity are less well characterized. Moreover, since different protocols are often used, comparing the effect of b-LED towards both microorganisms and epithelial surfaces may be difficult. The aim of this study was to analyze, in the same experimental setting, both the bactericidal activity and the effects on human keratinocytes. Exposure to b-LED induced an intense cytocidal activity against Gram-positive (i.e, *Staphylococcus aureus*) and Gram-negative (i.e., *Pseudomonas aeruginosa*) bacteria associated with catheter-related infections. Treatment with b-LED of a human keratinocyte cell line induced a transient cell cycle arrest. At the molecular level, exposure to b-LED induced a transient downregulation of Cyclin D1 and an upregulation of p21, but not signs of apoptosis. Interestingly, a transient induction of phosphor-histone γ-H2Ax, which is associated with genotoxic damages, was observed. At the same time, keratinocytes underwent a transient epithelial to mesenchymal transition (EMT)-like phenotype, characterized by E-cadherin downregulation and SNAIL/SLUG induction. As a functional readout of EMT induction, a scratch assay was performed. Surprisingly, b-LED treatment provoked a delay in the scratch closure. In conclusion, we demonstrated that b-LED microbicidal activity is associated with complex responses in keratinocytes that certainly deserve further analysis.

## 1. Introduction

Antimicrobial resistance (AMR) is the capacity of a microorganism to acquire the ability to replicate in the presence of a concentration of an antimicrobial agent that would instead be generally sufficient to inhibit or inactivate microorganisms of the same species [1]. This is an increasing global health concern with a high epidemiological impact on the whole population (increase in morbidity and mortality), with heavy social and economic burdens (extension of hospital stays/greater use of diagnostic procedures).

A study by the European Center for Disease Prevention and Control (ECDC) reports that in 2015, in the countries of the European Union and the European Economic Area, there were 671,689 cases of AMR infections, to which 33,110 deaths were attributable, a third of which occurred in Italy, highlighting the seriousness of the problem in our country. Compared to non-resistant forms, resistant bacteria will double the chances of developing a serious health issue and triple the chances of death [2]. There is a huge economic impact for this phenomenon. In the US, more than two million infections a year are caused by bacteria that are resistant to at least first-line antibiotic treatments, implying for the US health system 20 billion USD in excess costs each year [3,4].

The development of AMR can occur in surprisingly short times. This phenomenon is favored by the rapid bacterial replication rate (one replication cycles every 20 min) and by the spontaneous mutations in bacterial genome. As long as the resistance is limited to a single class of drug, it is possible to use different classes of molecules to overcome the problem. However, multiple resistances often arise, which are identified with the acronyms “MDR” (multi-drug-resistant), “XDR” (extensively drug-resistant) and “PDR” (pan-drug-resistant) [5].

Sepsis is the second leading cause of death in chronic renal patients on hemodialysis/peritoneal dialysis (HD/DP) [6]. These data are particularly relevant given the significant and continuous growth of the population undergoing dialysis. According to the United States Renal Data System Report, at the end of 2018, 495,052 patients were undergoing hemodialysis, with an increase of 8.8% between 2017 and 2018. Peritoneal dialysis patients increased to 58,636, representing a 7.7% growth since 2017. Most patients perform dialysis through catheter devices (venous or peritoneal) [7].

The permanent interruption of the skin, due to the presence of central venous or peritoneal catheters, is responsible for the infections in the majority of cases with a significant clinical impact. The most important risk factors are the duration of catheter placement, skin colonization at the catheter insertion site, and the frequent manipulation of the venous or peritoneal line. Dialysis patients often carry all three of these factors [8].

Light-emitting diodes (LEDs) are an optoelectronic device that exploits the ability of some semiconductor materials to produce photons through a spontaneous emission phenomenon when crossed by an electric current [9]. Currently, LEDs are available at wavelengths ranging from ultraviolet (UV) to visible to near infrared (NIR) bandwidth (247 to 1300 nm) [9].

LED light does not deliver enough power to damage tissues and does not have the same risk of accidental eye damage that lasers do. Visible/NIR-LED therapy has been deemed to present no significant risk by the Food and Drug Administration and has been approved for use in humans [9]. However, the effects of LED light exposure on eukaryotic cell physiopathology, including keratinocyte cell plasticity, have so far been incompletely studied.

Some preliminary data show that LED treatments are effective for the inactivation of pathogenic microbes (Gram-positive bacteria, Gram-negative bacteria, mycobacteria). Specifically, blue LED (b-LED) in the spectrum of 400–470 nm seems to have intrinsic antimicrobial properties resulting from the presence of endogenous photosensitizing chromophores in pathogenic microbes [10,11]. Accordingly, Dungel et al. showed that treatment with both red and b-LED significantly enhanced angiogenesis and perfusion in a skin flap model in rats [12].

LED photo-biomodulation treatment has also been shown to accelerate the resolution of radiation-induced dermatitis in breast cancer patients. Patients with diffuse type rosacea, keratosis pilaris rubra, as well as post-intervention erythema can benefit from a quicker recovery with complementary LED therapy [13,14].

However, such studies are generally performed either on microbes or cellular/tissue compartments, without analyzing both aspects simultaneously.

The aim of the study was to evaluate the bactericidal power of b-LED on a Gram-positive bacterium (*Staphylococcus aureus*) and on a Gram-negative bacterium (*P**seudomonas aeruginosa*), which are often causes of catheter-related infections, analyzing the effects on a keratinocyte cell line in terms of cytotoxicity and cellular plasticity in the same setting.

We found that exposure to b-LED exerts an antimicrobial activity both on Gram-positive and Gram-negative microorganisms. Furthermore, it promotes a transient growth blockage in HaCaT cells with transient induction of a genotoxic marker, but not apoptosis. At the same time, transient epithelial to mesenchymal transition (EMT)-like features were induced. Surprisingly, scratch closure was impaired in cells exposed to b-LED.

These studies demonstrated that emissions causing bacterial killing may induce complex events in keratinocyte plasticity deserving of further investigation.

## 2. Results

### 2.1. Antimicrobial Activity of 420 nm b-LED

Irradiation of *P. aeruginosa* and *S. aureus* strains for 4 h with 420 nm b-LED light resulted in statistically significant bacterial growth suppression when compared with non-irradiated controls in both bacterial strains. Post b-LED irradiation surviving fractions of all strains are shown in Figure 1. Specifically, the logarithmic reduction of CFU in the b-LED treatment of *P. aeruginosa* was a ~2.1 Log_10_ CFU/mL, which is equivalent to a bacterial killing value of 99.2%. The logarithmic reduction of CFU in the b-LED-treated *S. aureus* was ~1.4 Log_10_ CFU/mL, which is a bacterial killing value of 96.3%. The results obtained with bacterial strains were confirmed using the clinical isolate *P. aeruginosa* 19,595 (~1.5 Log_10_ reduction in CFU, ~96.9% of killing). These results indicate a bactericidal activity of the 420 nm b-LED light device used in this study.

### 2.2. Exposure of HaCaT Cells to b-LED Induces a Transient Cell Cycle Arrest

We then analyzed the activity of 420 nm b-LED on HaCaT cells, a human keratinocyte cell line, using the same experimental setting as above.

Cell viability was first evaluated with a calcein AM assay. A limited reduction in cell viability (vital cells/total cells) was found upon 24 h exposure to b-LED (Figure 2A). The count of total cells suggested a more evident reduction of cell proliferation at 24 h after treatment (Figure 2B). Analysis of cleaved caspase 3 ruled out induction of apoptosis after b-LED treatment (Figure 2C).

Changes in proliferation rate were confirmed by a cell proliferation assay. Exposure to b-LED reduced cell proliferation at 24 h post irradiation (Figure 2D). However, 48 h after irradiation, cell proliferation rate was restored to levels comparable to the control group.

We then analyzed molecular markers of cell cycle progression. p21 was analyzed as a readout of cell cycle blockage, whereas Cyclin D1 was considered a readout of cell cycle progression [15,16]. Exposure to b-LED induced a significant upregulation of p21 (*p* < 0.05), peaking at 4 h and returning to basal levels at 24 h post-irradiation (Figure 2E). Accordingly, Cyclin D1 (*p* < 0.001) was significantly inhibited at 4 h and was re-expressed at normal levels 24 h after irradiation (Figure 2F). These results were confirmed by WB analysis (Figure 2G,H). These results suggest that exposure of HaCaT cells to b-LED induces a transient cell cycle arrest, which is rescued at 24 h post irradiation.

### 2.3. Exposure of HaCaT Cells to b-LED Induces a Transient Phosphorylation of Histone γ-H2AX

Moreover, we investigated the expression of histone γ-H2AX as a readout of genotoxic damage [17]. b-LED radiation was compared with treatment with etoposide, a chemotherapeutic drug inducing double-strand breaks [18]. Exposure to b-LED induced a transient expression of phospho-γ-H2AX, which was reverted at 24 h after exposure, as shown by both WB (Figure 3A,B) and immunofluorescence (Figure 3C,D). On the other hand, etoposide promoted an expression of phospho-γ-H2AX histone, which was still persistent at 48 h after exposure.

### 2.4. Exposure of HaCaT Cells to b-LED Induces a Transient EMT-like Molecular Features with Delay in Wound Closure

To analyze whether exposure to b-LED induces EMT in HaCaT cells, we analyzed the expression of SNAIL and SLUG, two master genes of EMT, and direct E-Cadherin repressors. The expression of both SNAIL and SLUG, analyzed by qPCR, was significantly increased at 4 h post b-LED exposure. Among these two transcription factors, SNAIL expression was more intense and persistent than SLUG, being still increased at 48 h from irradiation compared to untreated controls (Figure 4A). At the same time, the expression of E-cadherin was persistently repressed (Figure 4A). E-cadherin and SLUG expression were confirmed by WB (Figure 4B). We then performed a scratch assay as a measure of directed migration, which is a functional readout of EMT. Unexpectedly, a scratch assay performed 24 and 48 h after b-LED radiation showed a delay in the migration of b-LED-irradiated HaCaT cells with respect to those of control group (Figure 4C,D). However, full wound closure occurred in both groups of HaCaT cells after 48 h.

## 3. Discussion

There is a growing interest in alternative approaches to mitigate skin infections. This need is strong in the case of patients carrying catheters, such as the peritoneal dialysis population, since they are often subject to chronic infections that are difficult to eradicate with standard antibiotic therapy [19].

Although light in the ultraviolet (UV) spectrum (300–400 nm) is used to treat various skin diseases such as psoriasis, atopic dermatitis (eczema), and vitiligo, it causes DNA adducts that have been linked to skin cancers and premature photoaging [20,21,22].

Visible light in the 400–760 nm range is presumably not associated with harmful DNA adducts, and may represent a safer alternative to UV phototherapy. However, the biological effects, underlying mechanisms, and clinical uses of different wavelengths of visible light have still not been completely characterized.

Moreover, most of the studies do not apply “side-by-side” comparison between the effect of b-LED on microbes and that on host cells. Some studies have suggested the existence of a therapeutic window where microbes are selectively inactivated while host cells are preserved [23,24,25]. In this setting, irradiation with b-LED is a promising alternative approach.

Consistent with previous reports, our results clearly show that 420 nm b-LED has strong antimicrobial activity against *S. aureus* and *P. aeruginosa* (about 1.5–2 log reduction) [26].

A possible mechanism underlying the biological effect of b-LED is related to photochemical reactions involving the absorption of a specific wavelength of light by photoreceptor molecules, such as endogenous porphyrins. The absorption of light in the visible light spectrum between the wavelengths 400 and 500 nm may trigger the production of reactive oxygen species (ROS) such as singlet oxygen, the hydroxyl radical, and the superoxide anion that are detrimental to bacteria, thus producing the observed antimicrobial effect. Indeed, DNA manipulation to knockout porphyrin synthesis impact microbe sensitivity to b-LED [11,27].

While the bactericidal effect of b-LED has been widely characterized by other research groups, the effect on eukaryotic cells has rarely been investigated, especially in a human setting, to date.

It is commonly accepted that b-LED is much less harmful to host cells than UV irradiation [28]. The DNA damage caused by UV results in (i) misincorporation of bases during replication process, (ii) hydrolytic damage, which results in deamination of bases, depurination, and depyrimidination, and (iii) oxidative damage through ROS induction [29].

Instead, no evidence of b-LED genotoxicity was observed in mouse skin in vivo when subjected to b-LED therapeutic exposure to inactivating biofilms [30]. The hypothesized mechanism of the cytotoxic effect of b-LED on host cells is similar to that on bacteria, which is associated with the photo-excitation of intracellular chromophores sensitive to blue light and the subsequent generation of cytotoxic ROS [31,32].

Ideally, the b-LED wavelength used should selectively excite the chromophores in pathogenic bacteria, while the photo-excitation of chromophores in host cells should be minimal.

Investigating the effects of b-LED-mediated oxidative stress is useful, because ROS play a dual role. They are toxic byproducts of aerobic metabolism, but are also required for the progression of numerous basic biological processes including cell proliferation and differentiation [33,34,35]. Moreover, ROS may stimulate the expression of the transcription factor SNAIL, the EMT master gene [36,37]. Therefore, induction of EMT-fibrosis upon exposure to b-LED may be secondary to oxidative stress induced by ROS [38]. For this reason, the expression of EMT markers was evaluated in this experimental setting. In in vivo conditions, the induction of a transient EMT upon exposure to b-LED would favor the re-epithelization and the cicatrization processes at the site of catheter implantation.

Our data show a reduced cell proliferation at 24 h post b-LED. However, the replicative capacity was completely re-established at 48 h post b-LED exposure. Data on cell proliferation are in accordance with molecular analysis of cell cycle effectors.

b-LED irradiation induced a temporary cell cycle arrest in HaCaT cells, as evidenced by the increased activity and expression of p21 and the simultaneous reduction of CycD1. A normal cell cycle was re-established 24 h after exposure, as demonstrated by the simultaneous re-expression of CycD1 and inhibition of p21. CycD1 plays a major role in the positive regulation of G1 progression. Enforced overexpression of D type cyclins can shorten the G1 interval in cultured cells. Furthermore, CycD1 is negatively regulated by a family of Cyclin-dependent kinase inhibitor protein (CDKI) including p21. The increase in p21 levels inhibits CycD1, thus contributing to G1 arrest [16].

We also investigated E-cadherin expression in HaCaT cells after b-LED exposure as a readout of EMT induction. E-cadherin mediates the composition of adherens junctions, which are one of the key components of the epidermal barrier. E-cadherin expression is directly regulated by the transcription factor SNAIL [39]. SNAIL expression in different epithelial cells leads to a conversion towards a fibroblastic phenotype at the same time that E-cadherin expression is lost and migratory properties are acquired. Besides E-cadherin, SNAIL promotes the expression of genes involved in the control of motility and migration [37]. In this experimental setting, we found a significant increase in both SNAIL and SLUG mRNA expression upon exposure to b-LED. At a protein level, a marked increase of SLUG was found.

Several lines of evidence suggest an important role for SLUG in normal adult epidermis. SLUG expression promotes epithelial outgrowth of keratinocytes, including the EMT-like processes of wound healing [40,41,42]. One may hypothesize that ROS induced by exposure to b-LED have a role in SNAIL induction, as demonstrated by Radisky et al. [36].

It is necessary to point out that bHLH factors such as SNAIL and SLUG have additional cellular functions that sometimes occur independently of the induction of full EMT. The expression of these transcription factors may protect cells from the death induced either by the loss of survival factors or by direct apoptotic stimuli [43,44].

Therefore, SNAIL is a potent survival factor. SNAIL-expressing cells are resistant to the action of direct apoptotic stimuli and are resistant to DNA damage. Thus, these EMT-related transcription factors may not be able to mediate a full EMT in this experimental setting, but may be implicated in the induction of survival pathways [45,46,47].

As a functional readout of EMT induction, a scratch assay was performed. The scratch assay test showed an initial delay in the closure of the wound in the irradiated HaCaT cells that was filled at 48 h post exposure, with complete wound closure also in the irradiated b-LED cells.

It has been reported by previous studies that lasers favor wound healing; however, there is a lack of consensus on standardized treatment parameters such as wavelengths, dose, and therapeutic outcomes [48,49]. For this purpose, our study showing a reduction in scratch closure upon exposure to b-light may sound as a warning to stress the urgence of setting the optimal parameters in order to achieve an effective wound closure in pathological conditions.

Collectively, these findings support the conclusion that exposure to b-LED irradiation induces a partial and transient EMT, but this is not sufficient to induce a stable mesenchymal phenotype. Many lines of evidence focus now on the physiopatological effects of partial EMT (pEMT), or hybrid epithelial/mesenchymal phenotype both in tumor an non-tumor systems [50].

In conclusion, the resistance of HaCaT cells to b-LED, together with the bactericidal efficacy, makes this a promising innovative therapeutic approach. Other studies must be performed in order to demonstrate the safety and the possible utility of b-LED in the field of exit-site infection in dialysis and beyond.

## 4. Materials and Methods

### 4.1. Antibodies and Chemicals

Mouse monoclonal antibody against cleaved caspase 3 was from Cell Signaling Technology (Danvers, MA, USA); against E-cadherin was from BD (Franklin Lakes, NJ, USA); against tubulin was from Millipore (Merck, Kenilworth, NJ, USA); rabbit polyclonal antibodies against p21 and cyclin D1 were from Santa Cruz Biotech (Dallas, TX, USA); against SLUG was from Cell Signaling Technology; rabbit monoclonal antibody to γH2A.X (phospho S139) was from Abcam, (Cambridge, UK). Etoposide was from Sigma-Aldrich (Saint Louis, MO, USA).

### 4.2. b-LED Source

A device was designed for the present study consisting of 54 b-LED strips anchored on an aluminum surface and suitably sized to ensure a homogeneous surface radiance of 160 mW/cm^2^ and a wavelength of 420 nm. The radiant dose used in the study was chosen based on the results of preliminary studies on bactericidal efficacy and plausible non-cytotoxicity.

The overall dimensions of the device were 24 cm × 16.5 cm; the size of the radiation surface was 9 cm × 8 cm. The device was placed at a distance of 5 cm from the radiation field in all experiments to maintain the characteristics mentioned. The irradiance was controlled by adjusting the distance of the aperture of the LED and the target with the use of a power/energy meter. The irradiation time used was 4 h, similar to that of a standard dialysis session.

The LED device (Omika, Los Angeles, CA, USA model SMD 3528-300LED) was designed in Rome, Italy. The density of radiant energy expressed in units of joules per square centimeter (J/cm^2^) over time (hours, h) was equal to 237 J/cm^2^/h of delivered b-LED.

### 4.3. Bacterial Strains and Keratinocytes

The reference strains *S. aureus* ATCC 25923 and *P. aeruginosa* ATCC 27853 were used in this study. The clinical strain *P. aeruginosa* 19595 was isolated from a leg injury by the microbiology team of Policlinico Umberto I (Sapienza University of Rome) and was shown to be resistant to different antibiotics (i.e., cefepime, ceftazidime, ciprofloxacin and piperacillin/tazobactam).

Bacteria were cultured in Luria-Bertani (LB) medium and incubated at 37 °C until reaching an optical density (O.D.) of 0.8 at 590 nm, measured with an UV-1700 Pharma Spec spectrophotometer (Shimadzu, Tokyo, Japan). Afterwards, bacterial cells were centrifuged at 1400× *g* for 10 min and resuspended in phosphate buffered saline (PBS) at a final concentration of 1 × 10^6^ colony forming unit (CFU)/mL, and 500 µL were added onto individual wells in 24-well microplates.

Plates were left open in a biological safety cabinet class II and irradiated for 240 min with b-LED. Control (non-irradiated) cultures were not exposed to light. After 240 min, aliquots of 10 μL from control or treated samples were diluted 1:100 in PBS or directly spread onto LB-agar plates, respectively, for colony counting after an overnight incubation at 37 °C. Bactericidal activity was expressed as reduction in the number of CFU and in percentage with respect to the controls. Values represent the mean ± standard deviation (SD) of three independent experiments.

HaCaT cells were from (from AddexBio, San Diego, CA, USA). Cells were cultured in Dulbecco’s modified Eagle’s medium supplemented with 10% fetal calf serum, 50 U/mL penicillin, 50 μg/mL streptomycin.

### 4.4. Real Time Quantitative Polymerase Chain Reaction (qPCR)

To evaluate gene expression of HaCaT keratinocytes qPCR was performed in the treated and control groups. Results obtained with microarray and quantitative real-time PCR (qPCR) were then compared. qPCR quantitative real-time PCR of Cyclin d1, p21, SNAIL, SLUG, E-cadherin (ECAD), smooth muscle alpha-actin (ACTA 2) was performed. Total RNA, was extracted from cell cultures with ReliaPrep™ RNA Tissue Miniprep System (Promega, Madison, WI, USA) and quantified with NanoDrop™ 2000/2000c Spectrophotometers. An amount of 1 µg of total extract was reverse transcribed with iScriptTM c-DNA Synthesis Kit (Bio-Rad Laboratories, Inc., Hercules, CA, USA) according to the manufacturer’s instructions. cDNA was diluted and 20 ng were amplified by qPCR reaction using GoTaq^®^ qPCR Master Mix (Promega, La Jolla, CA, USA). The following specific primer pairs were used: for L34: 5′GTCCCGAACCCCTGGTAATAG3’ and 5′GGCCCTGCTGACATGTTTCTT3′; for SNAIL: 5′CACTATGCCGCGCTCTTTC3′ and 5′GCTGGAAGGTAAACTCTGGATTAGA3′; for E-cadherin: 5′TACGCCTGGGACTCCACCTA3′ and 5′CCAGAAACGGAGGCCTGAT3′; for SLUG: 5′TGGGCAAAGAACTACTGCG3′ and 5′AGAGTTGGCGGAGCTAAACAG3′; for p21: 5′GAGGAGGCGCCATGTCAGAA3′ and 5′AGTCACCCTCCAGTGGTGTC3′; for Cyclin D1: 5′CCTCTAAGATGAAGGAGACCA3′ and 5′CACTTGAGCTTGTTCACCA3′. Relative expression levels were calculated with the 2^(−ΔΔCt)^ method and were normalized to L34 ribosomal RNA.

### 4.5. Cell Viability Assay

HaCaT cells were plated at the concentration of 3 × 10 ^5^ cells per well in a 6-well plate. The next day, cells were incubated for 1 h in Hanks Balanced Salt Solution, HBSS (Gibco™, Life Technologies, Carlsbad, CA, USA) with 2 μM calcein AM (Invitrogen, Waltham, MA, USA), at 37 ° C in humidified 5% CO_2_ atmosphere. Cells were then left untreated or were exposed to b-LED for 4 h. Cell were then evaluated for cell viability assay at time 0, 24 and 48 h after b-LED exposure using calcein AM according to the manufacturer’s protocol. Images were acquired using a fluorescence microscope (Nikon Inverted Microscope Eclipse TE200, Amsterdam, The Netherlands). The experiment was performed in triplicate and repeated twice.

### 4.6. Cell Proliferation Assay

HaCaT cells were plated at the concentration of 2 × 10^4^ per plate in a 96-well plate. The next day, cells were left untreated or were exposed to b-LED for 4 h or and were evaluated for cell proliferation assays using the CellTiter 96 Aqueous One Solution Cell Proliferation Assay system (Promega) according to the manufacturer’s instructions. Briefly, 20 × 10^3^ cells were plated into each well in a 96-well and 10 μL per well of CellTiter 96 AQueous One Solution reagent was added. After 1 h incubation in humidified 5% CO_2_ atmosphere, absorbance at 490 nm was measured at 4, 24 and 48 h time interval after exposure to b-LED using a SpectraMax 13 microplate reader. The experiment was performed in triplicate and was repeated twice.

### 4.7. Scratch Assay

HaCaT cells were seeded on Culture-Insert 2 Well in µ-Dish 35 mm from ibidi (Martinsried, Germany) and were allowed to reach 100% confluency. Cells were then treated with b-LED for 4 h. A scratch wound was created removing the culture-insert as in [51,52]. Cells were then fixed for 10 min with PFA 4%, stained with Rhodamine-phalloidin and Hoechst 33342 (Invitrogen), and then imaged by confocal microscopy. The experiment was performed in triplicate and was repeated twice.

Confocal images were acquired at the Olympus iX83 FluoView1200 laser scanning confocal microscope using an UPLSAPO10x2, NA 0.40 air objective (Shinjuku, Japan). Images were stitched using Olympus FluoView software.

### 4.8. Immunofluorescence

For the analysis of phospho-γ-2AX localization, HaCaT cells were fixed for 10 min with PFA 4%, in PBS and permeabilized with 0.2% Triton X-100 (Sigma-Aldrich) in PBS. Cy3-conjugated secondary antibody was from Jackson Immunoresearch (Philadelphia, PA, USA). Coverslips were mounted in Prolong Gold antifade (Life Technologies) and examined under a confocal microscope (Leica TCS SP2, Wetzlar, Germany). Digital images were acquired with the Leica software and the image adjustments and merging were performed by using the appropriated tools of ImageJ software. A minimum of 4 fields per sample (at least 120 total cells per total) from two independent experiments were analyzed.

### 4.9. Western Blotting

Cells were lysed in Laemmli buffer, and Western blotting was performed as previously described [53]. Monolayers of MCs were lysed in modified RIPA buffer containing: 50 mM Tris-HCl, pH 7.4; 1% NP-40; 0.1% SDS; 0.25% Nadeoxycholate; 150 mM NaCl; 1 mM EDTA; 1 mM EGTA; 1 mM PMSF; 1 μg/mL each of aprotinin, leupeptin and pepstatin; and 25 mM NaF (all from Sigma). Equal amounts of protein were resolved by SDS-PAGE. Proteins were transferred to PVDF membranes (Millipore, Bedford, VA, USA) and probed with antibodies using standard procedures. PVDF-bound antibodies were detected by chemiluminescence with ECL (Amersham Life Sciences, Little Chalfont, UK).

### 4.10. Statistical Analysis

Statistical significance was determined with a *t*-test using GraphPad Prism version 8.0 (La Jolla, CA, USA). Differences were considered significant at *, *p* < 0.05; **, *p* < 0.01; *** *p* < 0.001 and ns, not significant.

## Figures and Tables

**Figure 1 ijms-23-01896-f001:**
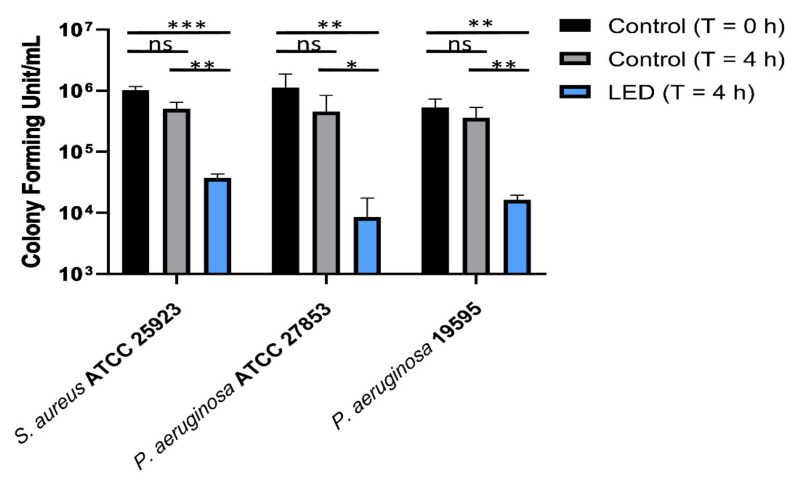
Number of viable cells of *S. aureus* ATCC 25923, *P. aeruginosa* ATCC 27853 and the clinical isolate *P. aeruginosa* 19595, after 4 h of treatment with b-LED (blue bars). Controls were untreated samples at time 0 (black bars) and 4 h (grey bars). The data reported are the mean ± standard deviation (SD) of three independent experiments. The level of statistical significance between samples was determined by the multiple *t* test (GraphPad Prism v.8.0.1, GraphPad Software, La Jolla, CA, USA), and indicated as follows: * *p* < 0.05; ** *p* < 0.01; *** *p* < 0.001 and ns, not significant.

**Figure 2 ijms-23-01896-f002:**
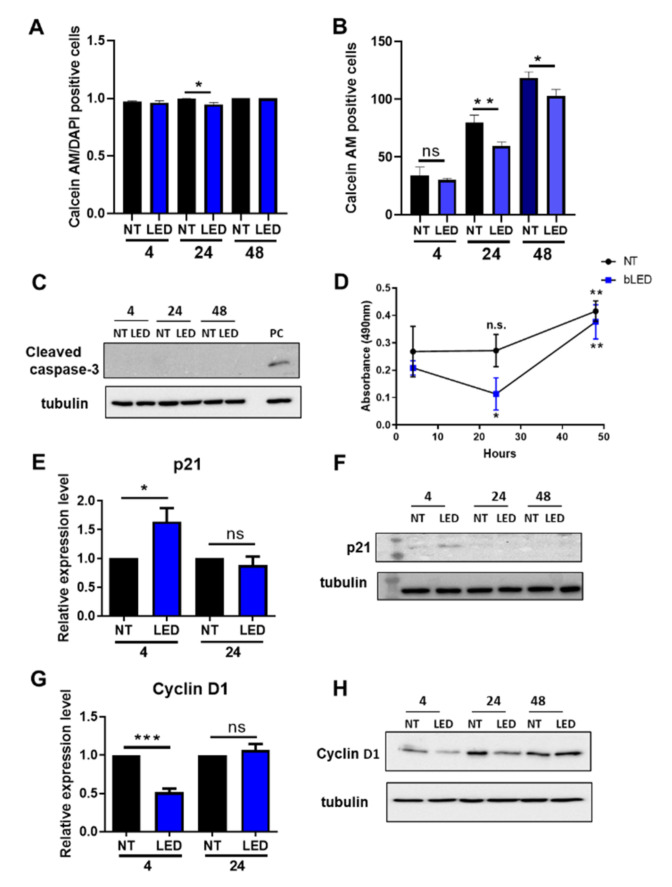
Exposure to b-LED causes a transient block in cell proliferation. (**A**,**B**) Cell viability test of HaCaT cells exposed to b-LED. HaCaT cells were pretreated with calcein AM (2 µM) and exposed to b-LED for 4 h. Cells were then analyzed 4, 24 and 48 h after beginning of exposure to b-LED. Cells were fixed and stained with DAPI. Images were acquired by fluorescence microscopy. (**A**) Calcein AM/DAPI-positive cells; (**B**) calcein AM-positive cells. Bars represent the mean ± SEM from two independent experiments. Twelve fields containing at least 30 nuclei per field were analyzed. * *p* < 0.05; ** *p* < 0.01. (**C**) Western blot showing the expression of cleaved caspase 3 in HaCaT cells exposed to b-LED for 4 h. Cells were analyzed 4, 24 and 48 h after beginning of exposure to b-LED. Western blot analysis was performed on total lysates. Tubulin was detected as a loading control. Data are representative of three independent experiments. (**D**) Cell proliferation assay of HaCaT cells exposed to b-LED. HaCaT cells were left untreated or were exposed to b-LED for 4 h and were then evaluated for cell proliferation assays 4, 24 and 48 h after beginning of exposure to b-LED using a SpectraMax 13 microplate reader. The experiment was performed in triplicate and was repeated twice. * *p* < 0.05; ** *p* < 0.01; ns, not significant. (**E**–**H**) HaCaT cells exposed to b-LED for 4 h and then were analyzed 4 and 24 h after beginning of exposure. (**E**) Quantitative RT-PCR expression analysis of p21 in. (**F**) Western blot expression of p21 in HaCaT cells. (**G**) quantitative RT-PCR expression analysis of Cyclin D1 in HaCaT cells. (**H**) Western blot expression of Cyclin D1 in HaCaT cells. Quantitative RT-PCR was performed on total RNA. L34 mRNA levels were used for normalization. Results are expressed in terms of fold change; bars represent the mean ± SEM of duplicate determinations in four independent experiments. * *p* < 0.05; *** *p* < 0.001 and ns, not significant. Western blot analysis was performed on total lysates. Tubulin was detected as a loading control. Data are representative of three independent experiments.

**Figure 3 ijms-23-01896-f003:**
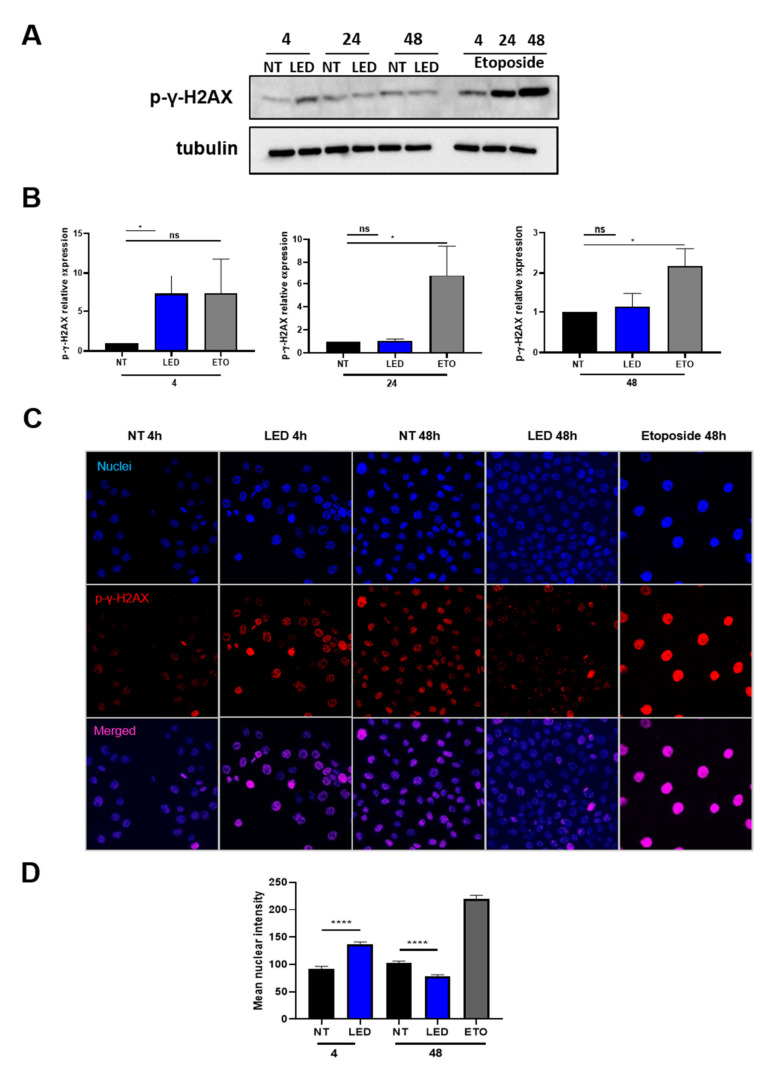
Exposure to b-LED causes a transient induction of γ -H2AX histone in HaCaT cells. (**A**) WB showing the expression of γ-H2AX histone (phospho S139) in HaCaT cells exposed to b-LED for 4 h and then analyzed at the indicated times. Etoposide (10 µM) was used as positive control. Western blot analysis was performed on total lysates. Tubulin was detected as a loading control. (**B**) Densitometric quantification of the experiment shown above. Results are expressed in terms of fold change. (**C**) Confocal immunofluorescence showing the expression of γ-H2AX histone (phospho S139) in cells stimulated as in (**A**). (**D**) quantification of the experiment shown in (**C**). At least 120 nuclei were quantified from 2 independent experiments. Bars represent the mean ± SEM of duplicate determinations in four independent experiments. * *p* < 0.05, **** *p* < 0.0001 and ns, not significant.

**Figure 4 ijms-23-01896-f004:**
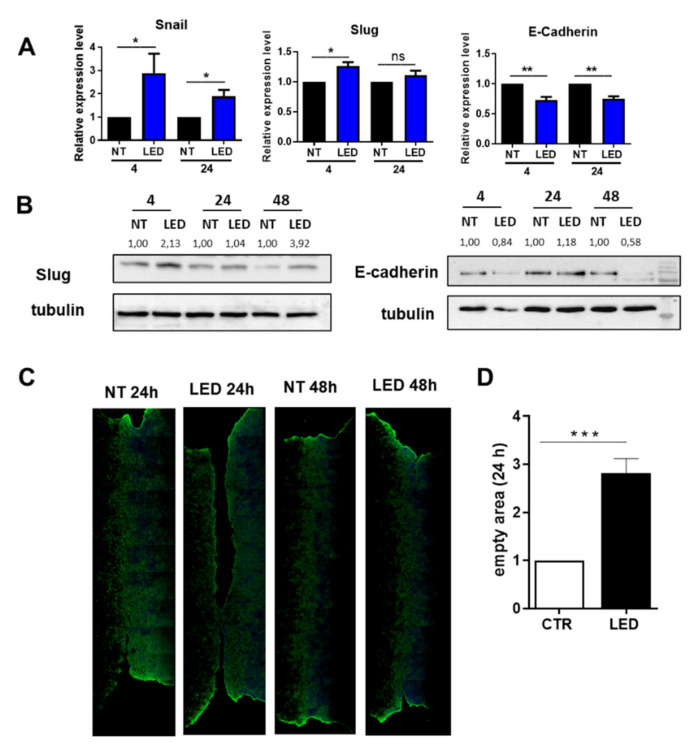
Exposure to b-LED causes a transient EMT phenotype with delay in wound closure. (**A**) Quantitative RT-PCR expression analysis of SNAIL, SLUG and E-cadherin in HaCaT cells exposed to b-LED for 4 h and then were analyzed 4 and 24 h after beginning of exposure. Quantitative RT-PCR was performed on total RNA. L34 mRNA levels were used for normalization. Results are expressed in term of fold change bars represent the mean ± SEM of duplicate determinations in at least five independent experiments. * *p* < 0.05; ** *p* < 0.01. (**B**) Western blot showing the expression of SLUG and E-cadherin in HaCaT cells exposed to b-LED for 4 h. Cells were analyzed 4, 24 and 48 h after the beginning of exposure to b-LED. Western blot analysis was performed on total lysates. Tubulin was detected as a loading control. Data are representative of three independent experiments. (**C**) Effect of exposure to b-LED on wound closure. HaCaT cells were left to reach 100% confluency in Ibidi μ-Dish plates. MCs were exposed to b-LED for 4 h. Then, 24 or 48 h after the beginning of exposure the insert was removed and after 18 h cells were fixed and stained with phalloidin (green) or Hoechst33342 (blue) to stain nuclei. Representative experiment is shown of three performed. (**D**) Quantification of the experiment shown in (**C**). Bars represent the mean ± SEM from three independent experiments. Two corresponding fields at time 0 and 24 h per experiments were measured. *** *p* < 0.001, ns, not significant.

## Data Availability

Not applicable.

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
