# Peer review of "Exposure to b-LED Light While Exerting Antimicrobial Activity on Gram-Negative and -Positive Bacteria Promotes Transient EMT-like Changes and Growth Arrest in Keratinocytes"

_ijms, 2022, doi:10.3390/ijms23031896_

Round 1
Reviewer 1 Report
The manuscript ijms-1582141 entitled “Exposure to b-LED light while exerting antimicrobial activity 2 on Gram-negative and -positive bacteria, promotes transient 3 EMT-like changes and growth arrest in keratinocytes” is a re-submission of recently submitted manuscript but with some improvements compared to previous one (e.g. use of etoposide as a PC).
The manuscript is scope of the journal, however there are some minor issues that need to be addressed.
- When presenting figures with multiple charts/blots, please mark every item. E.g. Figure 2 should be divided A-H. Please avoid terms ‘left’, ‘right’, ‘up’, ‘down’ and put letter for each one.
- How did you check for the normality of results, since you have used parametrical tests?
Please mind the minor errors such as:
uppercase F in Figure (line 118), use µ instead of u (line 361), use h for hours and correct throughout the manuscript (line 127-8, 135-150, 153-179, 350, etc.), et al. instead of and collaborators (line 298)
Author Response
Reviewer 1
The manuscript ijms-1582141 entitled “Exposure to b-LED light while exerting antimicrobial activity 2 on Gram-negative and -positive bacteria, promotes transient 3 EMT-like changes and growth arrest in keratinocytes” is a re-submission of recently submitted manuscript but with some improvements compared to previous one (e.g. use of etoposide as a PC).
The manuscript is scope of the journal, however there are some minor issues that need to be addressed.
- When presenting figures with multiple charts/blots, please mark every item. E.g. Figure 2 should be divided A-H. Please avoid terms ‘left’, ‘right’, ‘up’, ‘down’ and put letter for each one.
This has been dealt with in the text.
- How did you check for the normality of results, since you have used parametrical tests?
Please note that T-test has been applied for cell-based experiments. On the base of the limited numerosity of observations formally was not possible to confirm a normal distribution. However as previously reported, (see Furusawa et al.,Biophysics, doi: 10.2142/biophysics.1.25) cellular heterogeneity mostly relies on gaussian distribution.
Please mind the minor errors such as:
uppercase F in Figure (line 118), use µ instead of u (line 361), use h for hours and correct throughout the manuscript (line 127-8, 135-150, 153-179, 350, etc.), et al. instead of and collaborators (line 298)
This has been dealt with in the text.

Reviewer 2 Report
Review of the article: Exposure to b-LED light while exerting antimicrobial activity on Gram-negative and -positive bacteria, promotes transient EMT-like changes and growth arrest in keratinocytes.
Manuscript ID: ijms-1582141
In my opinion, the proposed manuscript is very interesting and well prepared. All experiments were well planned and performed. Below I have presented only several (minor of importance) suggestion and remarks that authors should take into account preparing the final version of the article.
Detailed comments
Abstract – No important critical remarks. In lines 28 and 29 the authors should write which species of Gram-positive and Gram-negative bacteria were investigated. The authors should also explain al abbreviations
Introduction – this part of manuscript is excellent.
Materials and methods
Lines 364-366 – some more details should be provided (volumes, dilutions etc.);
Lines 373-388 – some information about calculation of RT-PCR results should be provided
Results
Line 115 - Please explain why irradiation was performed for 4 hours – was it based on the results of some preliminary studies?
Line 118 – please use a capital letter for Figure 1
Figure 2D, Figure 3B and Figure 4A – in the legend to these figures some more information about result of RT-PCR should be presented – what exactly is shown on the Y axis
Discussion – well prepared no critical remarks.
Final decision – minor revision
Author Response
Reviewer 2
Abstract – No important critical remarks. In lines 28 and 29 the authors should write which species of Gram-positive and Gram-negative bacteria were investigated. The authors should also explain al abbreviations
Introduction – this part of manuscript is excellent.
Thank you for the consideration.
Materials and methods
Lines 364-366 – some more details should be provided (volumes, dilutions etc.);
This has been dealt with in the text.
Lines 373-388 – some information about calculation of RT-PCR results should be provided
We now provide more information in the Matherials and Methods section including how RT-PRC results were calculated.
Results
Line 115 - Please explain why irradiation was performed for 4 hours – was it based on the results of some preliminary studies?
We chose to expose keratinocytes to 4 hours of irradiation since this time may may correspond to a standard dialysis session. To use longer irradiation times would have limited the translational relevance of our discoveries.
Line 118 – please use a capital letter for Figure 1
This has been dealt with in the text.
Figure 2D, Figure 3B and Figure 4A – in the legend to these figures some more information about result of RT-PCR should be presented – what exactly is shown on the Y axis
Y axis shows the expression level of the specific gene analyzed in the figure (Specific gene/housekeeping) relative to NT (Fold-change). We now show in the Y axis ‘relative expression level’.

This manuscript is a resubmission of an earlier submission. The following is a list of the peer review reports and author responses from that submission.
Round 1
Reviewer 1 Report
This article starts with the antimicrobial activity of blue led light against P. aeruginosa and S. aureus. And then the author talks about the effect of b-led on cell toxicity, cell proliferation. Genotoxicity, epithelial-to-mesenchymal transition by observing E-cadherin, and performed scratch assay on HaCaT, a human keratinocytes cell line.
All this research has been previously done by different scientists all around the globe. This article repeats what has been done already. So I strongly disagree to publish this article in the “ molecular science journal”, I suggest submitting this article to “Applied science”.
Reviewer 2 Report
The manuscript ijms-1476844 entitled “Exposure to b-LED light while exerting antimicrobial activity on Gram-negative and –positive bacteria, promotes transient EMT-like changes and growth arrest in keratinocytes” describes the ability of blue LED light to exhibit micorbicidal properties, but also growth delay in skin cell lines.
The manuscript is scope of the journal, however there are some issues that need to be addressed.
- Please, add some data that would describe the severity of the AMR (lines 42-3). E.g. number of people affected, died of AMR sepsis, percentage of longer hospital stay, or increased costs, etc.
- Add a sentence (line 52) that would state how many people are in need of HD/HP procedures.
These data would contribute to making your research more relevant and to giving importance of a studied problem.
- Please write LED in full word upon its first mentioning (line 59).
- Can you add a sentence or two about the use of LED in similar (bio)medical treatments (paragraph 63-67), and perhaps to state if it has some microbicidal properties? It would better fit with the next sentence describing aims. Although it is a bit unusual to have a summary of results at the end of introduction, I actually liked this part.
- Add killing value in percentage for clinical isolate (line 90).
- Numerate Figures so that each one has its own letter. For Fig 2 use A-H, to avoid terms “A left”, “A right”.
- In Fig 2 “A right” looks like NT bar for 48 h should be black instead of purple.
- Is the choice of sodium arsenite a good choice for positive control for γ-H2AX assay, since DNA damage is ROS mediated and this is a marker of double-strand breaks? Perhaps etoposide or ionizing radiation would be better choice?
- Please move reference from line 202 to line 201 (“…reports (17), …”) as presently written it suggests that your results were already published in (17). Also, since you are referring to reports and there is only one reference, please add other reports.
- Can you relate and discuss the power used when comparing UV and b-LED?
- Since the wound closure is delayed, and besides the b-LED’s antimicrobial properties, could this feature of a treatment be considered as a major limitation in its application?
- Is this EMT-like physiology related to possible cancer induction or promotion?
- Please correct length (line 288), it seems LED strips were 420 nm long.
- Was this cytotoxicity tested towards same cell lines (line 291)?
- Please add a producer and a model of power/energy meter (line 296).
- How did you check for the normality of results, since you have used parametrical tests?
Please mind the minor errors such as:
numbers in keywords (line 36), use of γ in H2AX (line 30), capitalize figure (line 85), please be consistent and follow guides for authors (figure (line 85), Fig. 2A (line 105), Fig 2C (line 108), hours (line 108) or h (line 114) check entire manuscript), the use of capital letters for Snail and Slug – please check when you are referring to a gene and when to a protein, µ instead of u (line 308), space 0h (line 342), subscript CO2 (line 352).